# Meta-Learning Dynamics Forecasting Using Task Inference

**Rui Wang***
UC San Diego

**Robin Walters***
Northeastern University

**Rose Yu**
UC San Diego

## Abstract

Current deep learning models for dynamics forecasting struggle with generalization. They can only forecast in a specific domain and fail when applied to systems with different parameters, external forces, or boundary conditions. We propose a model-based meta-learning method called `DyAd` which can generalize across heterogeneous domains by partitioning them into different tasks. `DyAd` has two parts: an encoder which infers the time-invariant hidden features of the task with weak supervision, and a forecaster which learns the shared dynamics of the entire domain. The encoder adapts and controls the forecaster during inference using adaptive instance normalization and adaptive padding. We prove that the generalization error of such a procedure is governed by task relatedness in the source domain and domain differences between source and target. Experimentally, we demonstrate that our model outperforms state-of-the-art approaches to forecasting complex physical dynamics including turbulent flow, real-world sea surface temperature and ocean currents. Our code is available open sourced at `https://github.com/Rose-STL-Lab/Dynamic-Adaptation-Network`.

## 1 Introduction

Modeling dynamical systems with deep learning has shown great success in a wide range of systems from climate science, Internet of Things (IoT) to infectious diseases [23, 64, 8, 33, 42, 22]. However, the main limitation of previous works is limited generalizability. Most approaches only focus on a specific system and train on past data in order to predict the future. Thus, a new model must be trained to predict a system with different dynamics. Consider, for example, learning fluid dynamics; shown in Fig. 1 are two fluid flows with different degrees of turbulence. Although the flows are governed by the same equations, the difference in buoyant forces would require two separate deep learning models to forecast. Therefore, it is imperative to develop *generalizable* deep learning models for dynamical systems that can learn and predict well over a large heterogeneous domain.

Meta-learning [63, 7, 13], or learning to learn, improves generalization by learning multiple tasks from the environment. Recent developments in meta-learning have been successfully applied to few-shot classification [46, 62], active learning [77], and reinforcement learning [18, 20]. However, meta-learning in the context of forecasting high-dimensional physical dynamics has not been studied before. The challenges with meta-learning dynamical systems are unique in that (1) we need to efficiently infer the latent representation of the dynamical sys-

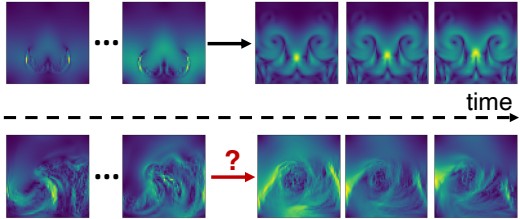

Figure 1: Meta-learning dynamic forecasting on turbulent flow. The model needs to generalize to a flow with a very different buoyant force.

tem given observed time series data, (2) we need to account for changes in unknown initial and boundary conditions, and (3) we need to model the temporal dynamics across heterogeneous domains.

36th Conference on Neural Information Processing Systems (NeurIPS 2022).

Our approach is inspired by the fact that similar dynamical systems may share time-invariant hidden features. Even the slightest change in these features may lead to vastly different phenomena. For example, in climate science, fluids are governed by a set of differential equations called Navier-Stokes equations. Some features such as kinematic viscosity and external forces (e.g. gravity), are time-invariant and determine the flow characteristics. By inferring this latent representation, we can model diverse system behaviors from smoothly flowing water to atmospheric turbulence.

Inspired by neural style transfer [26], we propose a model-based meta-learning method, called `DyAd`, which can rapidly adapt to systems with varying dynamics. `DyAd` has two parts, an encoder $g$ and a forecaster $f$. The encoder maps different dynamical systems to time-invariant hidden features representing constants of motion, boundary conditions, and external forces which characterize the system. The forecaster $f$ then takes the learnt hidden features from the encoder and the past system states to forecast the future system state. Controlled by the time-invariant hidden features, the forecaster has the flexibility to adapt to a wide range of systems with heterogeneous dynamics.

Unlike gradient-based meta-learning techniques such as MAML [13], `DyAd` automatically adapts during inference using an encoder and does not require retraining. Similar to model-based meta-learning methods such as MetaNets [46], we employ a two-part design with an adaptable learner which receives task-specific weights. However, for time-series forecasting, since input and output come from the same domain, a support set of labeled data is unnecessary to define the task. The encoder can infer the task directly from the query input.

Our contributions include:

- A novel model-based meta-learning method (`DyAd`) for dynamic forecasting in a large heterogeneous domain.

- An encoder capable of extracting the time-invariant hidden features of a dynamical system using time-shift invariant model structure and weak supervision.

- A new adaptive padding layer (AdaPad), designed for adapting to boundary conditions.

- Theoretical guarantees for `DyAd` on the generalization error of task inference in the source domain as well as domain adaptation to the target domain.

- Improved generalization performance on heterogeneous domains such as fluid flow and sea temperature forecasting, even to new tasks outside the training distribution.

## 2 Methods

### 2.1 Meta-learning in dynamics forecasting

Let $\mathbf{x} \in \mathbb{R}^d$ be a $d$-dimensional state of a dynamical system governed by parameters $\psi$. The problem of dynamics forecasting is that given a sequence of past states $(\mathbf{x}_1, \ldots, \mathbf{x}_t)$, we want to learn a map $f$ such that: $f : (\mathbf{x}_1, \ldots, \mathbf{x}_t) \longmapsto (\mathbf{x}_{t+1}, \ldots \mathbf{x}_{t+k})$.

Here $l$ is the length of the input series, and $h$ is the forecasting horizon in the output. Existing approaches for dynamics forecasting only predict future data for a specific system as a single task. Here a task refers to forecasting for a specific system with a given set of parameters. The resulting models often generalize poorly to different system dynamics. Thus a new model must be trained to predict for each specific system.

To perform meta-learning, we identify each forecasting task by some parameters $c \subset \psi$, such as constants of motion, external forces, and boundary conditions. We learn multiple tasks simultaneously and infer the task from data. Here we use $c$ for a subset of system parameters $\psi$, because we usually do not have the full knowledge of the system dynamics. In the turbulent flow example, the state $\mathbf{x}_t$ is the velocity field at time $t$. Parameters $c$ can represent Reynolds number, mean vorticity, mean magnitude, or a vector of all three.

Let $\mu$ be the data distribution over $\mathcal{X} \times \mathcal{Y}$ representing the function $f \colon \mathcal{X} \to \mathcal{Y}$ where $\mathcal{X} = \mathbb{R}^{d \times t}$ and $\mathcal{Y} = \mathbb{R}^{d \times k}$. Our main assumption is that the domain $\mathcal{X}$ can be partitioned into separate tasks $\mathcal{X} = \cup_{c \in \mathcal{C}} \mathcal{X}_c$, where $\mathcal{X}_c$ is the domain for task $c$ and $\mathcal{C}$ is the set of all tasks. The data in the same task share the same set of parameters. Let $\mu_c$ be the conditional distribution over $\mathcal{X}_c \times \mathcal{Y}$ for task $c$.

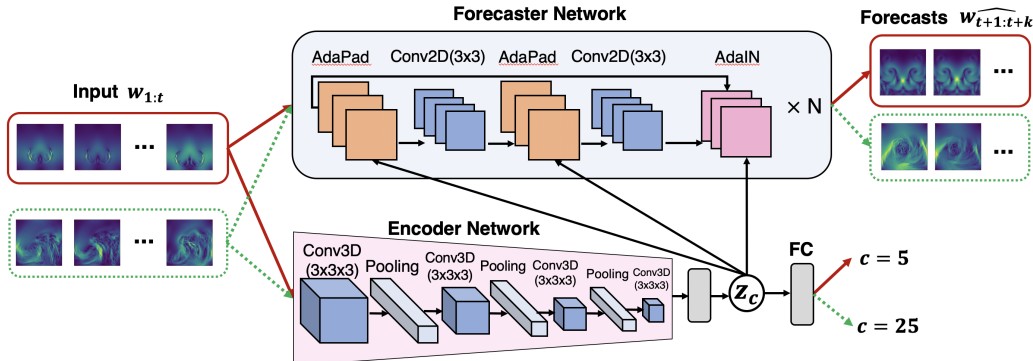

Figure 2: Overview of `DyAd` applied to two inputs of fluid turbulence, one with small external forces and one with larger external forces. The encoder infers the time-shift invariant characteristic variable $z$ which is used to adapt the forecaster network.

During training, the model is presented with data drawn from a subset of tasks $\{(x,y) : (x,y) \sim \mu_c, c \sim C\}$. Our goal is to learn the function $f \colon \mathcal{X} \to \mathcal{Y}$ over the whole domain $\mathcal{X}$ which can thus generalize across all tasks $c \in \mathcal{C}$. To do so, we need to learn the map $g \colon \mathcal{X} \to \mathcal{C}$ taking $x \in \mathcal{X}_c$ to $c$ in order to infer the task with minimal supervision.

## 2.2 `DyAd`: Dynamic Adaptation Network

We propose a model-based meta-learning approach for dynamics forecasting. Given multiple forecasting tasks, we propose to learn the function $f$ in two stages. That is, by first inferring the task $c$ from the input $x$, and then adapting to a specialized forecaster $f_c \colon \mathcal{X}_c \to \mathcal{Y}$ for each task. An alternative is to use a single deep neural network to directly model $f$ in one step over the whole domain. But this requires the training set to have good and uniform coverage of the different tasks. If the data distribution $\mu$ is highly heterogeneous or the training set is not sampled i.i.d. from the whole domain, then a single model may struggle with generalization.

We hypothesize that by partitioning the domain into different tasks, the model would learn to pick up task-specific features without requiring uniform coverage of the training data. Furthermore, by separating task inference and forecasting into two stages, we allow the forecaster to rapidly adapt to new tasks that never appeared in the training set.

As shown in Fig. 2, our model consists of two parts: an encoder $g$ and a forecaster $f$. We introduce $z_c$ as a time-invariant hidden feature for task $c$. We assume that $c$ depends linearly on the hidden feature for simplicity and easy interpretation. We design the encoder to infer the hidden feature $z_c$ given the input $x$. We then use $z_c$ to adapt the forecaster $f$ to the specific task, i.e., model $y = f_c(x)$ as $y = f(x, z_c)$. As the system dynamics are encoded in the input sequence $x$, we can feed the same input sequence $x$ to a forecaster and generate predictions $\hat{y} = f_c(x)$.

## 2.3 Encoder Network

The encoder maps the input $x$ to the hidden features $z_c$ that are time-invariant. To enforce this inductive bias, we encode time-invariance both in the architecture and in the training objective.

**Time-Invariant Encoder.** The encoder is implemented using 4 Conv 3D layers, each followed by `BatchNorm`, `LeakyReLU`, and max-pooling. Note that theoretically, max-pooling is not perfectly shift invariant since 2x2x2 max-pooling is equivariant to shifts of size 2 and only approximately invariant to shifts of size 1. But standard convolutional architectures often include max-pooling layers to boost performance. We convolve both across spatial and temporal dimensions.

After that, we use a global mean-pooling layer and a fully connected layer to estimate the hidden feature $\hat{z}_c$. The task parameter depends linearly on the hidden feature. We use a fully connected layer to compute the parameter estimate $\hat{c}$.

Since convolutions are equivariant to shift (up to boundary frames) and mean pooling is invariant to shift, the encoder is shift-invariant. In practice, shifting the time sequence one frame forward will add one new frame at the beginning and drop one frame at the end. This creates some change in output value of the encoder. Thus, practically, the encoder is only approximately shift-invariant.

**Encoder Training.** The encoder network $g$ is trained first. To combat the loss of shift invariance from the change from the boundary frames, we train the encoder using a time-invariant loss. Given two training samples $(x^{(i)}, y^{(i)})$ and $(x^{(j)}, y^{(j)})$ and their task parameters $c$, we have loss

$$\mathcal{L}_{\text{enc}} = \sum_{c \sim \mathcal{C}} \|\hat{c} - c\|^2 + \alpha \sum_{i,j,c} \|\hat{z}_c^{(i)} - \hat{z}_c^{(j)}\|^2 + \beta \sum_{i,c} \|\|\hat{z}_c^{(i)}\|^2 - m\|^2 \tag{1}$$

where $\hat{z}^{(i)} = g(x^{(i)})$ and $\hat{z}^{(j)} = g(x^{(j)})$ and $\hat{c}^{(i)} = W\hat{z}_c^{(i)} + b$ is an affine transformation of $z_c$.

The first term $\|\hat{c} - c\|^2$ uses weak supervision of the task parameters whenever they are available. Such weak supervision helps guide the learning of hidden feature $z_c$ for each task. While not all parameters of the dynamical system are known, we can compute approximate values in the datum $c^{(i)}$ based on our domain knowledge. For example, instead of the Reynolds number of the fluid flow, we can use the average vorticity as a surrogate for task parameters.

The second term $\|\hat{z}_c^{(i)} - \hat{z}_c^{(j)}\|^2$ is the time-shift invariance loss, which penalizes the changes in latent variables between samples from different time steps. Since the time-shift invariance of convolution is only approximate, this loss term drives the time-shift error even lower. The third term $\|\|\hat{z}_c^{(i)}\| - m|^2$ ($m$ is a positive value) prevents the encoder from generating small $\hat{z}_c^{(i)}$ due to time-shift invariance loss. It also helps the encoder to learn more interesting $z$, even in the absence of weak supervision.

**Hidden Features.** The encoder learns time-invariant hidden features. These hidden features resemble the time-invariant dimensionless parameters [30] in physical modeling, such as Reynolds number in fluid mechanics.

The hidden features may also be viewed as partial disentanglement of the system state. As suggested by [37, 47], our disentanglement method is guided by inductive bias and training objectives. Unlike complete disentanglement, as in e.g. [39], in which the latent representation is factored into time-invariant and time-varying components, we focus only on time-shift-invariance.

Nonetheless, the hidden features can control the forecaster which is useful for generalization.

## 2.4 Forecaster Network.

The forecaster incorporates the hidden feature $z_c$ from the encoder and adapts to the specific forecasting task $f_c = f(\cdot, z_c)$. In what follows, we use $z$ for $z_c$. We use two specialized layers, adaptive instance normalization (AdaIN) and adaptive padding (AdaPad). AdaIN has been used in neural style transfer [26, 21] to control generative networks. Here, AdaIN may adapt for specific coefficients and external forces. We also introduce a new layer, $\text{AdaPad}(x, z)$, which is designed for encoding the boundary conditions of dynamical systems. The backbone of the forecaster network can be any sequence prediction model. We use a design that is similar to `ResNet` for spatiotemporal sequences.

**AdaIN.** We employ AdaIN to adapt the forecaster network. Denote the channels of input $x$ by $x_i$ and let $\mu(x_i)$ and $\sigma(x_i)$ be the mean and standard deviation of channel $i$. For each AdaIN layer, a particular style is computed $s = (\mu_i, \sigma_i)_i = Az + b$, where the linear map $A$ and bias $b$ are learned weights. Adaptive instance normalization is then defined as $y_i = \sigma_i \frac{x_i - \mu(x_i)}{\sigma(x_i)} + \mu_i$. In essence, the channels are renormalized to the style $s$.

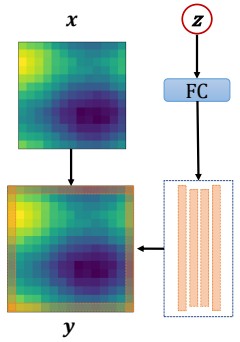

For dynamics forecasting, the hidden feature $z$ encodes data analogous to the coefficients of a differential equation and external forces on the system. In numerical simulation of a differential equation these coefficients enter as scalings of different terms in the equation and the external forces are added to the combined force equation [24]. Thus in our context AdaIN, which scales channels and adds a global vector, is well-suited to injecting this information.

Figure 3: Illustration of the AdaPad operation.

**AdaPad.** To complement $\mathrm{AdaIN}$, we introduce $\mathrm{AdaPad}$, which encodes the boundary conditions of each specific dynamical system. Generally when predicting dynamical systems, error is introduced along the boundaries since it is unknown how the dynamics interact with the boundary of the domain, and there may be unknown inflows or outflows. In our method, the inferred hidden feature $z$ may contain the boundary information. $\mathrm{AdaPad}$ uses the hidden features to compute the boundary conditions via a linear layer. Then it applies the boundary conditions as padding immediately outside the spatial domain in each layer, as shown in Fig. 3.

**Forecaster Training** The forecaster is trained separately after the encoder. The kernels of the convolutions and the mappings of the $\mathrm{AdaIN}$ and $\mathrm{AdaPad}$ layers are all trained simultaneously as the forecaster network is trained. Denote the true state as $y$ and the predicted state as $\hat{y}$, we compute the loss per time step $\|\hat{y} - y\|^2$ for each example. We accumulate the loss over different time steps and generate multi-step forecasts in an autoregressive fashion. In practice, we observe separate training achieves better performances than training end-to-end, see experiments for details.

### 2.5 Theoretical Analysis

The high-level idea of our method is to learn a good representation of the dynamics that generalizes well across a heterogeneous domain, and then adapt this representation to make predictions on new tasks. Our model achieves this by learning on multiple tasks simultaneously and then adapting to new tasks with domain transfer. We prove that learning the tasks simultaneously as opposed to independently results in better generalization (Proposition B.3 in Appendix). We also provide a theoretical decomposition for the generalization error.

Specifically, our hypothesis space has the form $\{x \mapsto f_\theta(x, g_\phi(x))\}$ where $\phi$ and $\theta$ are the weights of the encoder and forecaster respectively. Let $\epsilon_{\mathcal{X}}$ be the error over the entire domain $\mathcal{X}$, that is, for all $c$. Let $\epsilon_{\mathrm{enc}}(g_\phi) = \mathbb{E}_{x \sim \mathcal{X}}(\mathcal{L}_1(g(x), g_\phi(x))$ be the encoder error where $g\colon \mathcal{X} \to \mathcal{C}$ is the ground truth. Let $\mathcal{G} = \{g_\phi \colon \mathcal{X} \to \mathcal{C}\}$ be the encoder hypothesis space. Denote the empirical risk of $g_\phi$ by $\hat{\epsilon}_{\mathrm{enc}}(g_\phi)$. $W_1$ denotes the Wasserstein distance between tasks. $R(\mathcal{G})$ and $R(\mathcal{F})$ represent the Rademacher complexities of encoder and forecaster. The following Proposition decomposes the generalization error in terms of forecaster error, encoder error, and distance between tasks.

**Proposition 2.1.** *Assume the forecaster $c \mapsto f_\theta(\cdot, c)$ is Lipschitz continuous with Lipschitz constant $\gamma$ uniformly in $\theta$ and $l \leq 1/2$. Let $\lambda_c = \min_{f \in \mathcal{F}}(\epsilon_c(f) + 1/K \sum_{k=1}^{K} \epsilon_{c_k}(f))$. For large $n$ and probability $\geq 1 - \delta$,*

$$\epsilon_{\mathcal{X}}(f_\theta(\cdot, g_\phi(\cdot))) \leq \gamma \hat{\epsilon}_{\mathrm{enc}}(g_\phi) + \frac{1}{K}\sum_{k=1}^{K} \hat{\epsilon}_{c_k}(f_\theta(\cdot, c_k)) + \mathbb{E}_{c \sim \mathcal{C}}\left[ W_1\left(\hat{\mu}_c, \frac{1}{K}\sum_{k=1}^{K}\hat{\mu}_{c_k}\right) + \lambda_c \right]$$
$$+ 2\gamma R(\mathcal{G}) + 2R(\mathcal{F}) + (\gamma + 1)\sqrt{\log(1/\delta)/(2nK)} + \sqrt{2\log(1/\delta)}\left(\sqrt{1/n} + \sqrt{1/(nK)}\right).$$

This result helps to quantify the trade-offs with respect to different terms in the error bound in our two-part architecture and the error of the model can be controlled by minimizing the empirical errors of the encoder and forecaster. See the Appendix Proposition B.6 for full proofs.

## 3 Related Work

**Learning Dynamical Systems.** Deep learning models are gaining popularity for learning dynamical systems [60, 8, 29, 5, 73, 64, 51]. An emerging topic is physics-informed deep learning [53, 6, 11, 69, 4, 3, 68, 12, 10] which integrates inductive biases from physical systems to improve learning. For example, [45, 6] incorporated Koopman theory into the architecture. [43] used deep neural networks to solve PDEs with physical laws enforced in the loss functions. [17] and [9] build models upon Hamiltonian and Lagrangian mechanics that respect conservation laws. [68] proposed a hybrid approach by marrying two well-established turbulent flow simulation techniques with deep learning to produce a better prediction of turbulence. [10] propose a physics-informed GAN architecture that uses physics knowledge to inform the learning of both the generator and discriminator models. However, these approaches deal with a *specific* system dynamics instead of a large *heterogeneous* domain.

**Multi-task learning and Meta-learning.** Multi-task learning [65] focuses on learning shared representations from multiple related tasks. Architecture-based MTL methods can be categorized into encoder-focused [35] and decoder-focused [74]. There are also optimization-based MTL methods, such as task balancing methods [27]. But MTL assumes tasks are known a priori instead of inferring the task from data. On the other hand, the aim of meta-learning [63] is to leverage the shared representation to fast adapt to unseen tasks. Based on how the meta-level knowledge is extracted and used, meta-learning methods are classified into model-based [46, 1, 50, 59, 78], metric-based [67, 61] and gradient-based [13, 57, 16, 76]. Most meta-learning approaches are not designed for forecasting with a few exceptions. [72] proposed to train a domain classifier with weak supervision to help domain adaptation but focused on low-dimensioal time series classification problems. [50] designed a residual architecture for time series forecasting with a meta-learning parallel. [1] proposed a modular meta-learning approach for continuous control. But forecasting physical dynamics poses unique challenges to meta-learning as it requires encoding physical knowledge into our model.

**Style Transfer.** Our approach is inspired by neural style transfer techniques. In style transfer, a generative network is controlled by an external style vector through adaptive instance normalization between convolutional layers. Our hidden representation bears affinity with the "style" vector in style transfer techniques. Rather than aesthetic style in images, our hidden representation encodes time-invariant features. Style transfer initially appear in non-photorealistic rendering [31]. Recently, neural style transfer [25] has been applied to image synthesis [15, 36], videos generation [56], and language translation [52]. For dynamical systems, [58] adapts texture synthesis to transfer the style of turbulence for animation. [28] studies unsupervised generative modeling of turbulent flows but for super-resolution reconstruction rather than forecasting.

**Video Prediction.** Our work is also related to video prediction. Conditioning on the historic observed frames, video prediction models are trained to predict future frames, e.g., [41, 14, 75, 66, 49, 14, 70, 71, 32, 40, 34]. There is also conditional video prediction [48] which achieves controlled synthesis. Many of these models are trained on natural videos from unknown physical processes. Our work is substantially different because we do not attempt to predict object or camera motions. However, our method can be potentially combined with video prediction models to improve generalization.

# 4 Experiments

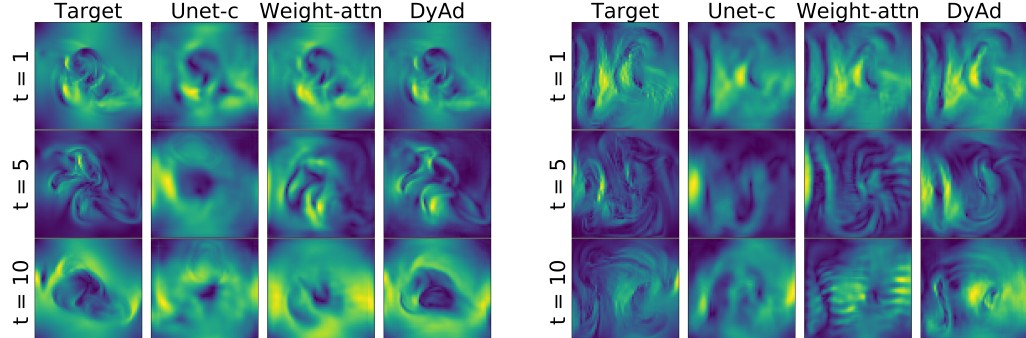

Figure 4: Target and predictions by `Unet-c`, `Modular-wt` and `DyAd` at time 1, 5, 10 for turbulent flows with buoyancy factors 9 (left) and 21 (right) respectively. `DyAd` can easily generate predictions for various flows while baselines have trouble understanding and disentangling buoyancy factors.

## 4.1 Datasets

We experiment on three datasets: synthetic turbulent flows, real-world sea surface temperature and ocean currents data. These are difficult to forecast using numerical methods due to unknown external forces and complex dynamics not fully captured by simplified mathematical models.

**Turbulent Flow with Varying Buoyancy.** We generate a synthetic dataset of turbulent flows with a numerical simulator, PhiFlow[1]. It contains $64 \times 64$ velocity fields of turbulent flows in which we vary

---
[1] https://github.com/tum-pbs/PhiFlow

Table 1: Prediction RMSE on the turbulent flow and sea surface temperature datasets. Prediction RMSE and ESE (energy spectrum errors) on the future and domain test sets of ocean currents dataset.

| Model | Turbulent Flows | | Sea Temperature | | Ocean Currents | |
|---|---|---|---|---|---|---|
| | future | domain | future | domain | future | domain |
| ResNet | $0.94\pm0.10$ | $0.65\pm0.02$ | $0.73\pm0.14$ | $0.71\pm0.16$ | $9.44\pm1.55$ \| $0.99\pm0.15$ | $9.65\pm0.16$ \| $0.90\pm0.16$ |
| ResNet-c | $0.88\pm0.03$ | $0.64\pm0.01$ | $0.70\pm0.08$ | $0.71\pm0.06$ | $9.71\pm0.01$ \| $0.81\pm0.03$ | $9.15\pm0.01$ \| $0.73\pm0.03$ |
| U-Net | $0.92\pm0.02$ | $0.68\pm0.02$ | $0.57\pm0.05$ | $0.63\pm0.05$ | $7.64\pm0.05$ \| $0.83\pm0.02$ | $7.61\pm0.14$ \| $0.86\pm0.03$ |
| Unet-c | $0.86\pm0.07$ | $0.68\pm0.03$ | $0.47\pm0.02$ | $0.45\pm0.06$ | $\mathbf{7.26}\pm\mathbf{0.01}$ \| $0.94\pm0.02$ | $7.51\pm0.03$ \| $0.87\pm0.04$ |
| PredRNN | $0.75\pm0.02$ | $0.75\pm0.01$ | $0.67\pm0.12$ | $0.99\pm0.07$ | $8.49\pm0.01$ \| $1.27\pm0.02$ | $8.99\pm0.03$ \| $1.69\pm0.01$ |
| VarSepNet | $0.67\pm0.05$ | $0.63\pm0.06$ | $0.63\pm0.14$ | $0.49\pm0.09$ | $9.36\pm0.02$ \| $0.63\pm0.04$ | $7.10\pm0.01$ \| $0.58\pm0.02$ |
| Mod-attn | $0.63\pm0.12$ | $0.92\pm0.03$ | $0.89\pm0.22$ | $0.98\pm0.17$ | $8.08\pm0.07$ \| $0.76\pm0.11$ | $8.31\pm0.19$ \| $0.88\pm0.14$ |
| Mod-wt | $0.58\pm0.03$ | $0.60\pm0.07$ | $0.65\pm0.08$ | $0.64\pm0.09$ | $10.1\pm0.12$ \| $1.19\pm0.72$ | $8.11\pm0.19$ \| $0.82\pm0.19$ |
| MetaNet | $0.76\pm0.13$ | $0.76\pm0.08$ | $0.84\pm0.16$ | $0.82\pm0.09$ | $10.9\pm0.52$ \| $1.15\pm0.18$ | $11.2\pm0.16$ \| $1.08\pm0.21$ |
| MAML | $0.63\pm0.01$ | $0.68\pm0.02$ | $0.90\pm0.17$ | $0.67\pm0.04$ | $10.1\pm0.21$ \| $0.85\pm0.06$ | $10.9\pm0.79$ \| $0.99\pm0.14$ |
| DyAd+ResNet | $\mathbf{0.42}\pm\mathbf{0.01}$ | $\mathbf{0.51}\pm\mathbf{0.02}$ | $0.42\pm0.03$ | $0.44\pm0.04$ | $7.28\pm0.09$ \| $\mathbf{0.58}\pm\mathbf{0.02}$ | $\mathbf{7.04}\pm\mathbf{0.04}$ \| $\mathbf{0.54}\pm\mathbf{0.03}$ |
| DyAd+Unet | $0.58\pm0.01$ | $0.59\pm0.01$ | $\mathbf{0.35}\pm\mathbf{0.03}$ | $\mathbf{0.42}\pm\mathbf{0.05}$ | $7.38\pm0.01$ \| $0.70\pm0.04$ | $7.46\pm0.02$ \| $0.70\pm0.07$ |

the buoyant force acting on the fluid from 1 to 25. Each buoyant force corresponds to a forecasting task and there are 25 tasks in total. We use the mean vorticity of each task as partial supervision $c$ as we can directly calculate it from the data. Vorticity can characterize formation and circular motion of turbulent flows.

**Sea Surface Temperature.** We evaluate on a real-world sea surface temperature data generated by the NEMO ocean engine [38][2]. We select an area from Pacific ocean range from 01/01/2018 to 12/31/2020. The corresponding latitude and longitude are (-150~-120, -20~-50). This area is then divided into 25 64×64 subregions, each is a task since the mean temperature varies a lot along longitude and latitude. For the encoder training, we use season as an additional supervision signal besides the mean temperature of each subregion. In other words, the encoder should be able to infer the mean temperature of the subregion as well as to classify four seasons given the temperature series.

**Ocean Currents.** We also experiment with the velocity fields of ocean currents from the same region and use the same task division as the sea surface temperature data set. Similar to the turbulent flow data set, we use the mean vorticity of each subregion as the weak-supervision signal.

## 4.2 Baselines

We include several SoTA baselines from meta-learning and dynamics forecasting.

- ResNet [19]: A widely adopted video prediction model [69].
- U-net [55]: Originally developed for biomedical image segmentation, adapted for dynamics forecasting [11].
- ResNet/Unet-c: Above ResNet and Unet with an additional final layer that generates task parameter $c$ and trained with weak-supervision and forecasting loss altogether.
- PredRNN [70]: A state-of-art RNNs-based spatiotemporal forecasting model.
- VarSepNet [12]: A convolutional dynamics forecasting model based on spatiotemporal disentanglement.
- Mod-attn[1]: A modular meta-learning method which combines the outputs of modules to generalize to new tasks using attention.
- Mod-wt: A modular meta-learning variant which uses attention weights to combine the parameters of the convolutional kernels in different modules for new tasks.
- MetaNet [46]: A model-based meta-learning method which requires a few samples from test tasks as a support set to adapt.
- MAML [13]: A optimization-based meta-learning approach. We replaced the original classifier with a ResNet for regression.

---

[2]The data are available at `https://resources.marine.copernicus.eu/?option=com_csw&view=details&product_id=GLOBAL_ANALYSIS_FORECAST_PHY_001_024`

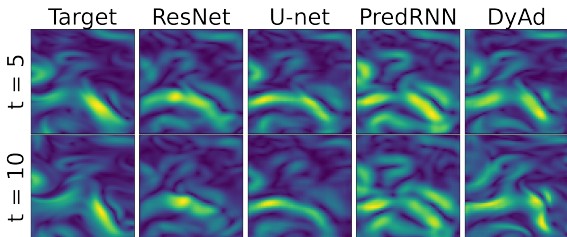

| Model | future | domain |
|-------|--------|--------|
| DyAd(ours) | **0.42**±**0.01** | **0.51**±**0.02** |
| No_enc | 0.63±0.03 | 0.60±0.02 |
| No_AdaPad | 0.47±0.01 | 0.54±0.02 |
| Wrong_enc | 0.66±0.02 | 0.62±0.03 |
| End2End | 0.45±0.01 | 0.54±0.01 |

Table 2: Ablation study: prediction RMSE of DyAd and its variations with different components removed from DyAd.

Figure 5: DyAd, ResNet, U-net, PredRNN velocity norm ($\sqrt{u^2 + v^2}$) predictions on an ocean current sample in the future test set.

Note that both ResNet-c and Unet-c have access to task parameters $c$. Modular-attn has a convolutional encoder $f$ that takes the same input $x$ as each module $M$ to generate attention weights, $\sum_{l=1}^{m} \frac{\exp[f(x)(l)]}{\sum_{k=1}^{m} \exp[f(x)(k)]} M_l(x)$. Modular-wt also has the same encoder but to generate weights for combining the convolution parameters of all modules. MetaNet requires samples from test tasks as a support set and MAML needs adaptation retraining on test tasks, while other models do not need any information from the test domains. Thus, we use additional samples of up to 20% of the test set from test tasks. MetaNet uses these as a support set. MAML is retrained on these samples for 10 epoch for adaptation. To demonstrate the generalizability of DyAd, we experimented with ResNet and U-net as our forecaster.

### 4.3 Experiments Setup

For all datasets, we use a sliding-window approach to generate samples of sequences. We evaluate on two scenarios of generalization. For test-future, we train and test on the same task but different time steps. For test-domain, we train and test on different tasks with an 80-20 split. All models are trained to make next step prediction given the history. We forecast in an autoregressive manner to generate multi-step ahead predictions. All results are averaged over 3 runs with random initialization.

Apart from the root mean square error (RMSE), we also report the energy spectrum error (ESE) for ocean current prediction, which quantifies the physical consistency. ESE indicates whether the predictions preserve the correct statistical distribution and obey the energy conservation law, which is a critical metric for physical consistency. See details about energy spectrum and complete experiments details in Appendix A.1.

## 5 Results

### 5.1 Prediction Performance.

Table 1 shows the RMSE of multi-step predictions on Turbulent Flows (20 steps), Sea Surface Temperature (10 steps), and Ocean Currents (10 step) in two testing scenarios. We observe that DyAd makes the most accurate predictions in both scenarios across all datasets. Comparing ResNet/Unet-c with DyAd, we observe the clear advantage of task inference with separate training. VarSepNet achieves competitive performances on Ocean Currents (second best) through spatiotemporal disentanglement but cannot adapt to future data. Table 1 also reports ESEs on real-world Ocean Currents. DyAd not only has small RMSE but also obtains the smallest ESE, suggesting it captures the statistical distribution of ocean currents well.

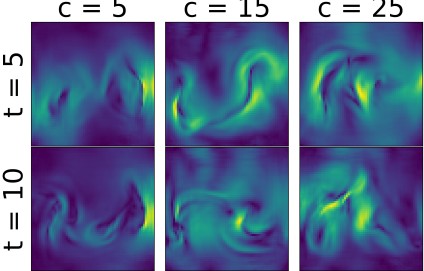

Figure 6: Outputs from DyAd while we vary encoder input but keep the forecaster input fixed. From left to right, the encoder is fed with flow with different buoyancy factor $c = 5, 15, 25$. the forecaster network input has fixed buoyancy $c = 15$.

Figure 4 shows the target and the predicted velocity norm fields ($\sqrt{u^2 + v^2}$) by Unet-c, Modular-wt and DyAd at time step 1, 5, 10 for Turbulent Flows with buoyancy factors 9 and 21 respectively. We can see that DyAd can generate realistic flows with the corresponding characteristics while the

baselines have trouble understanding and disentangling the buoyancy factor. Figure 5 shows `DyAd`, `ResNet`, `U-net`, `PredRNN` predictions on an ocean current sample in the future test set, and we see the shape of predictions by `DyAd` is closest to the target. These results demonstrate that `DyAd` not only forecasts well but also accurately captures the physical characteristics of the system.

## 5.2 Controllable Forecast.

`DyAd` infers the hidden features from data, which allows direct control of the latent space in the forecaster. We vary the encoder input while keeping the forecaster input fixed. Figure 6 right shows the forecasts from `DyAd` when the encoder is fed with flows having different buoyancy factors $c = 5, 15, 25$. As expected, with higher buoyancy factors, the predictions from the forecaster become more turbulent. This demonstrates that the encoder can successfully disentangle the latent representation of difference tasks, and control the predictions of the forecaster.

## 5.3 Ablation Study.

We performed several ablation studies of `DyAd` on the turbulence dataset to understand the contribution of each model component.

**Ablation of the Model and Training Strategy.** We first performed ablation study of the model architecture, shown in Table 2. We first remove the encoder from `DyAd` while keeping the same forecaster network (`No_enc`). The resulting model degrades but still outperforms `ResNet`. This demonstrates the effectiveness of `AdaIN` and `AdaPad` for forecasting. We also tested `DyAd` with AdaIN only (`No_AdaPad`), and the performance without AdaPad was slightly worse.

Another notable feature of our model is the ability to infer tasks with weakly supervised signals $c$. It is important to have a $c$ that is related to the task domain. As an ablative study, we fed the encoder in `DyAd` with a random $c$, leading to `Wrong_enc`. We can see that having the totally wrong supervision may slightly hurt the forecasting performance. We also trained the encoder and the forecaster in `DyAd` altogether (`End2End`) but observed worse performance. This validates our hypothesis about the significance of domain partitioning and separate training strategy.

**Ablation of Encoder Training Losses.** We performed an ablation study of three training loss terms for the encoder to show the necessity of each loss term, shown in Table 3 We tried training the encoder only with the supervision of task parameters (`Sup_c`). We also tested training with the first two terms but without the magnitude loss (`Sup_c+time_inv`). We can see that missing any of three loss terms would make the encoder fail to learn the correct task-specific and time-invariant features.

**Alternatives to AdaIN.** We also tested 5 different alternatives to AdaIN for injecting the hidden feature $z_c$ into the forecaster, and reported the results in Table 4. We tested concatenating $z_c$ to the input of the forecaster, using $z_c$ as the kernels of the forecaster, concatenating $z_c$ to the hidden states, adding $z_c$ to the hidden states, using $z_c$ as the biases in the convolutional layers in the forecaster. AdaIN worked better than any alternative we tested.

| Model | future | domain |
|---|---|---|
| DyAd (ours) | **0.42** | **0.51** |
| Sup_c | 0.53 | 0.56 |
| Sup_c+time_inv | 0.51 | 0.54 |

Table 3: Ablation study of the encoder training loss terms. We tried training the encoder only with the weak supervision (`Sup_c`), without the magnitude loss (`Sup_c+time_inv`).

| RMSE | AdaIN | ConI | KGen | ConH | Sum | Bias |
|---|---|---|---|---|---|---|
| future | **0.42** | 1.09 | 0.78 | 0.75 | 0.84 | 0.84 |
| domain | **0.51** | 0.85 | 0.78 | 0.74 | 0.80 | 0.78 |

Table 4: We tested concatenating $z_c$ to the input of the forecaster (ConI), using $z_c$ as the kernels of the forecaster (KGen), concatenating $z_c$ to the hidden states (ConH), adding $z_c$ to the hidden states(Sum), using $z_c$ as the biases in the convolutional layers (Bias). AdaIN worked better than any alternative we tested.

## 6 Conclusion

We propose a model-based meta-learning method, `DyAd` to forecast physical dynamics. `DyAd` uses an encoder to infer the parameters of the task and a prediction network to adapt and forecast giving the inferred task. Our model can also leverage any weak supervision signals that can help distinguish

different tasks, allowing the incorporation of additional domain knowledge. On challenging turbulent flow prediction and real-world ocean temperature and currents forecasting tasks, we observe superior performance of our model across heterogeneous dynamics. Future work would consider non-grid data such as flows on a graph or a sphere.

## Acknowledgments and Disclosure of Funding

We are grateful to several anonymous reviewers for their comments which have helped us to improve this work. This work was supported in part by U.S. Department Of Energy, Office of Science grant DE-SC0022255, U. S. Army Research Office grant W911NF-20-1-0334, and NSF grants #2134274 and #2146343. R. Walters was supported by the Roux Institute and the Harold Alfond Foundation and NSF grants #2107256 and #2134178. This research used resources of the National Energy Research Scientific Computing Center, a DOE Office of Science User Facility supported by the Office of Science of the U.S. Department of Energy under Contract No. DE-AC02-05CH11231 using NERSC award ßASCR-ERCAP0022715.

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
