# A    Additional Results

## A.1    Experiments Details

We use a sliding window approach to generate samples of sequences. For test-future, we train and test on the same tasks but different time steps. For test-domain, we train and test on different tasks with an 80%-20% split. All models are trained to make next step prediction given the previous steps as input. We forecast in an autoregressive manner to generate multi-step ahead predictions. All results are averaged over 3 runs with random initialization.

We tuned learning rate (1e-2$\sim$1e-5), batch size (16$\sim$64), the number of unrolled prediction steps for backpropogation (2$\sim$6), and hidden size (64$\sim$512). We fixed the number of historic input frames as 20. The models that use ResNet, including `MAML`, `MetaNets` and `DyAd+ResNet`, have same number of residual blocks for fair comparison. When we trained the encoder on turbulence and sea surface temperature, we used $\alpha = 1$ and $\beta = 1$ in the Equ. 1. For ocean currents, we used $\alpha = 0.2$ and $\beta = 0.2$. We performed all experiments on 4 V100 GPUs. Table 5 in Appendix A displays the number of parameters of each fine-tuned model on the turbulence dataset.

We compare our model with a series of baselines on the multi-step forecasting with different dynamics. We consider two testing scenarios: (1) dynamics with different initial conditions (test-future) and (2) dynamics with different parameters such as external force (test-domain). The first scenario evaluates the models' ability to extrapolate into the future for the same task. The second scenario is to estimate the capability of the models to generalize across different tasks.

Apart from the root mean square error (RMSE), we also report the energy spectrum error (ESE) for ocean current prediction which quantifies the physical consistency. ESE indicates whether the predictions preserve the correct statistical distribution and obey the energy conservation law, which is a critical metric for physical consistency. The turbulence kinetic energy spectrum $E(k)$ is related to the mean turbulence kinetic energy as

$$\int_0^\infty E(k)dk = (\overline{(u')^2} + \overline{(v')^2})/2, \ \ \overline{(u')^2} = \frac{1}{T}\sum_{t=0}^T (u(t) - \bar{u})^2.$$

where the $k$ is the wavenumber and $t$ is the time step. The spectrum can describe the transfer of energy from large scales of motion to the small scales and provides a representation of the dependence of energy on frequency. Thus, the ESE can indicate whether the predictions preserve the correct statistical distribution and obey the energy conservation law. A trivial example that can illustrate why we need ESE is that if a model simply outputs moving averages of input frames, the accumulated RMSE of predictions might not be high but the ESE would be really big because all the small or even medium eddies are smoothed out.

## A.2    Addtional Experiments Results

Table 5: The number of parameters of the fine-tuned model of each architecture on the turbulent flow dataset

| ResNet | U-net | PredRNN | VarSepNet | Mod-attn | Mod-wt | MetaNets | MAML |
|--------|-------|---------|-----------|----------|--------|----------|------|
| 20.32  | 9.69  | 27.83   | 9.85      | 13.19    | 13.19  | 9.63     | 20.32 |

| DyAd+ResNet | DyAd+Unet | DyAd+Unet-Big |
|-------------|-----------|---------------|
| 15.60       | 12.20     | 23.64         |

# B    Theoretical Analysis

The high-level idea of our method is to learn a good representation of the dynamics that generalizes well across a heterogeneous domain, and then adapt this representation to make predictions on new tasks. Our model achieves this by learning on multiple tasks simultaneously and then adapting to new tasks with domain transfer. We prove that learning the tasks simultaneously as opposed

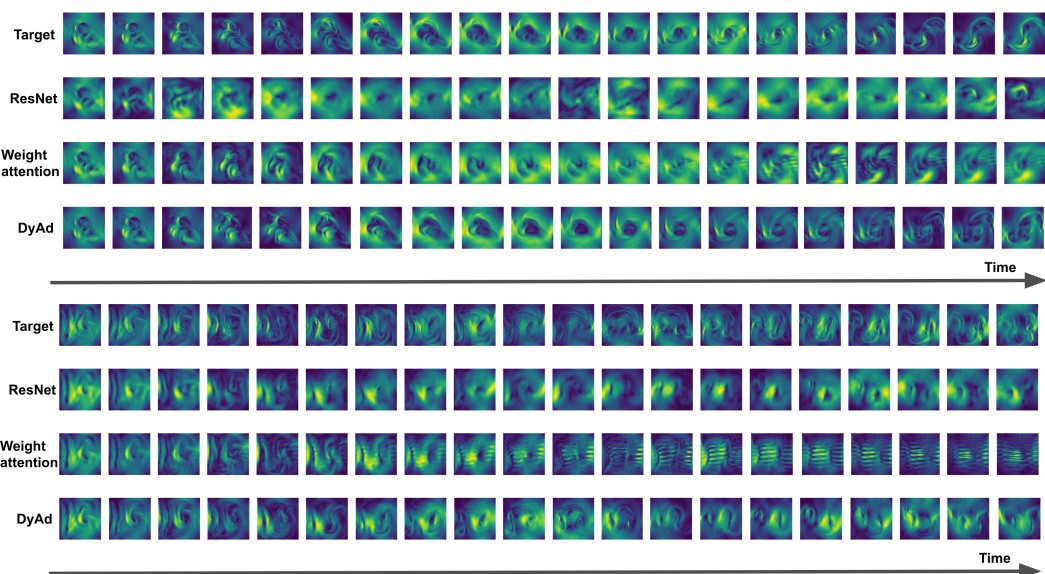

Figure 7: Target and full 20-step predictions by `Unet-c`, `Modular-wt` and `DyAd` turbulent flows with buoyancy factors 9 (top) and 21 (bottom) respectively.

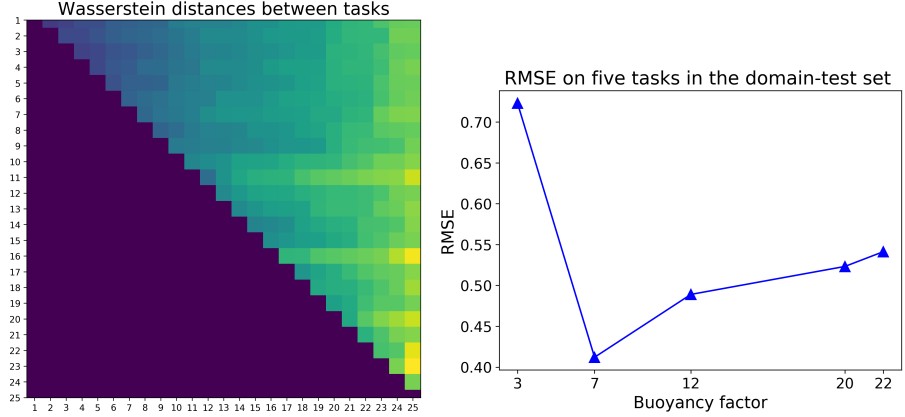

Figure 8: Left: Pairwise RMSEs between the averaged samples of different tasks in the turbulent flow dataset. RMSE between the averaged samples is a lower bound of Wasserstein distance between tasks. Right: DyAd+ResNet prediction RMSE breakdown on five tasks in the domain test set.

to independently results in tighter generalization bounds. As these are upper bounds, they do not necessarily imply better generalization, although empirically, this is what we observe. Although we are not able to provide a proof of our empirical finding that an encoder-forecaster architecture outperforms a monolithic architecture, we do provide an analysis of the error, decomposing it into error due to the forecaster, encoder, and domain adaptation. This decomposition helps to quantify the trade-offs involved in our two-part architecture.

Suppose we have $K$ tasks $\{c_k\}_{k=1}^K \sim \mathcal{C}$, each of which is sampled from a continuous parameter space $\mathcal{C}$. Here $c_k$ are the parameters of the task, which can be inferred by the encoder. For each task $c_k$, we have a collection of data $\hat{\mu}_{c_k}$ of size $n$, sampled from $\mu_k$, a shorthand for $\mu_{c_k}$.

## B.1 Multi-Task Learning Error

We want to bound the true loss $\epsilon$ using the empirical loss $\hat{\epsilon}$ and Rademacher complexity of the hypothesis class $\mathcal{F}$. We can use the classic results from [2]. Define empirical Rademacher complexity

for all tasks as

$$\hat{R}_{\mathbf{X}}(\mathcal{F}) = \mathbb{E}_\sigma \left[ \sup_{f \in \mathcal{F}} \left( \frac{1}{nK} \sum_{k=1}^K \sum_{i=1}^n \sigma_k^{(i)} l(f(\mathbf{x}_k^{(i)})) \right) \right] \qquad (2)$$

where $\{\sigma_k^{(i)}\}$ are independent binary variables $\sigma_k^{(i)} \in \{-1, 1\}$. The true Rademacher complexity is then defined $R(\mathcal{F}) = \mathbb{E}_{\mathbf{X}}(\hat{R}_{\mathbf{X}}(\mathcal{F}))$.

The following theorem restates the main result from [2]. We simplify the statement by using Rademacher complexity rather than the set cover number argument used in the original proof. The assumption of bounded loss is reasonable given a prescribed data domain and data preprocessing.

**Theorem B.1.** *([2]) Assume the loss is bounded $l \leq 1/2$. Given $n$ samples each from $K$ different forecasting tasks $\mu_1, \cdots, \mu_k$, then with probability at least $1 - \delta$, the following inequality holds for each $f \in \mathcal{F}$ in the hypothesis class:*

$$\epsilon(f) \leq \hat{\epsilon}(f) + 2R(\mathcal{F}) + \sqrt{\log(1/\delta)/(2nK)}$$

*Proof.* Consider $\{\mathbf{x}_k^{(i)}\}$ as independent random variables. For a function $\phi$ that satisfies

$$|\phi(\mathbf{x}^{(1)}, \cdots, \mathbf{x}^{(i)}, \cdots \mathbf{x}^{(n)}) - \phi(\mathbf{x}^{(1)}, \cdots, \tilde{\mathbf{x}}^{(i)}, \cdots \mathbf{x}^{(n)})| \leq c_i$$

by McDiarmid's inequality, we have

$$p\left( \phi(\mathbf{x}^{(1)}, \cdots, \mathbf{x}^{(n)}) - \mathbb{E}[\phi] \geq t \right) \leq \exp\left( -\frac{2t^2}{\sum_i c_i^2} \right).$$

Applying this inequality to the max difference $Q(\mathbf{X}) = \sup_{f \in \mathcal{F}}[\epsilon(f) - \hat{\epsilon}(f, \mathbf{X})]$, then with probability at least $1 - \delta$, we have

$$Q(\mathbf{X}) - \mathbb{E}_{\mathbf{X}}[Q(\mathbf{X})] \leq C\sqrt{\frac{\log 1/\delta}{nK}}$$

where $C$ is a constant depending on the bounds $c_i$. If the loss $l \leq 1/2$, then $|Q| \leq 1/2$ and so we can take $c_i = 1$ leading to $C = 1/\sqrt{2}$. A standard computation (see [44] Theorem 3.3) using the law of total expectation shows $\mathbb{E}_{\mathbf{X}}[Q(\mathbf{X})] \leq 2R(\mathcal{F})$, which finishes the proof. $\qquad \square$

The following inequality compares the performance for multi-task learning to learning individual tasks. Let $R_k(\mathcal{F})$ be the Rademacher complexity for $\mathcal{F}$ over $\mu_k$.

**Lemma B.2.** *The Rademacher complexity for multi-task learning is bounded $R(\mathcal{F}) \leq (1/K) \sum_{k=1}^K R_k(\mathcal{F})$.*

*Proof.* We compute the empirical Rademacher complexity,

$$\begin{aligned} \hat{R}_{\mathbf{X}}(\mathcal{F}) &= \mathbb{E}_\sigma \left[ \sup_{f \in \mathcal{F}} \left( \frac{1}{nK} \sum_{k=1}^K \sum_{i=1}^n \sigma_k^{(i)} l\left( f\left( \mathbf{x}_k^{(i)} \right) \right) \right) \right] \\ &\leq \mathbb{E}_\sigma \left[ \sum_{k=1}^K \sup_{f \in \mathcal{F}} \left( \frac{1}{nK} \sum_{i=1}^n \sigma_k^{(i)} l\left( f\left( \mathbf{x}_k^{(i)} \right) \right) \right) \right] \\ &= \frac{1}{K} \sum_{k=1}^K \mathbb{E}_{\sigma_k} \left[ \sup_{f \in \mathcal{F}} \left( \frac{1}{n} \sum_{i=1}^n \sigma_k^{(i)} l\left( f\left( \mathbf{x}_k^{(i)} \right) \right) \right) \right] \\ &= \frac{1}{K} \sum_{k=1}^K \hat{R}_{\mathbf{X}_k}(\mathcal{F}) \end{aligned}$$

$\qquad \square$

**Proposition B.3.** *Assume $n = n_k$ for all tasks $k$ and the loss $l$ is bounded $l \leq 1/2$, then the generalization bound given by considering each task individually is*

$$\epsilon(f) \leq \hat{\epsilon}(f) + 2\left( \frac{1}{K} \sum_{k=1}^K R_k(\mathcal{F}) \right) + \sqrt{\frac{\log 1/\delta}{2n}}, \qquad (3)$$

*which is weaker than the bound of Theorem B.1.*

*Proof.* Applying the classical analog of Theorem B.1 to a single task, we find with probability greater than $1 - \delta$,

$$\epsilon_k(f) \leq \hat{\epsilon}_k(f) + 2R_k(\mathcal{F}) + C_k \sqrt{\frac{\log 1/\delta}{n}}.$$

Averaging over all tasks yields

$$\frac{1}{K} \sum_{k=1}^{K} \epsilon_k(f) \leq \frac{1}{K} \sum_{k=1}^{K} \hat{\epsilon}_k(f) + 2 \frac{1}{K} \sum_{k=1}^{K} R_k(\mathcal{F}) + \frac{1}{K} \sum_{k=1}^{K} C_k \sqrt{\frac{\log 1/\delta}{n}}.$$

Since the loss $l$ is bounded $l \leq 1/2$, we can take $C = C_k = 1/\sqrt{2}$, giving the generalization upper bound for the joint error of Equation 3. By Lemma B.2 and the fact $1/\sqrt{2nK} \leq 1/\sqrt{2n}$, the bound in Theorem B.1 is strictly tighter. $\square$

The upper bound in Theorem B.1 is strictly tighter than that of Proposition B.3 as the first terms $\hat{\epsilon}(f)$ are equal, the second term is smaller by Lemma B.2 and the third is smaller since $1/\sqrt{2nK} \leq 1/\sqrt{2n}$. This helps explain why our multitask learning forecaster has better generalization than learning each task independently. The shared data tightens the generalization bound. Ultimately, though a tighter upper bound suggests lower error, it does not strictly imply it. We further verify this theory in our experiments by comparison to baselines which learn each task independently.

## B.2 Domain Adaptation Error

Since we test on $c \sim \mathcal{C}$ outside the training set $\{c_k\}$, we incur error due to domain adaptation from the source domains $\mu_{c_1}, \ldots, \mu_{c_K}$ to target domain $\mu_c$ with $\mu$ being the true distribution. Denote the corresponding empirical distributions of $n$ samples per task by $\hat{\mu}_c = \frac{1}{n_c} \sum_{i=1}^{n_c} \delta_{\mathbf{x}_c^{(i)}}$.

For different $c$ and $c'$, the domains $\mu_c$ and $\mu_{c'}$ may have largely disjoint support, leading to very high KL divergence. However, if $c$ and $c'$ are close, samples $\mathbf{x}_c \sim \mu_c$ and $\mathbf{x}_{c'} \sim \mu_{c'}$ may be close in the domain $\mathcal{X}$ with respect to the metric $\|\cdot\|_{\mathcal{X}}$. For example, if the external forces $c$ and $c'$ are close, given $\mathbf{x}_c \sim \mu_c$ there is likely $\mathbf{x}_{c'} \sim \mu_{c'}$ such that the distance between the velocity fields $\|\mathbf{x}_c - \mathbf{x}_{c'}\|$ is small. This implies the distributions $\mu_c$ and $\mu_{c'}$ may be be close in Wasserstein distance $W_1(\mu_c, \mu_{c'})$. The bound from [54] applies well to our setting as such:

**Theorem B.4.** *Let* $\lambda_c = \min_{f \in \mathcal{F}} \left( \epsilon_c(f) + 1/K \sum_{k=1}^{K} \epsilon_{c_k}(f) \right)$. *There is* $N = N(\dim(\mathcal{X}))$ *such that for* $n > N$, *for any hypothesis* $f$, *with probability at least* $1 - \delta$,

$$\epsilon_c(f) \leq \frac{1}{K} \sum_{k=1}^{K} \epsilon_{c_k}(f) + W_1 \left( \hat{\mu}_c, \frac{1}{K} \sum_{k=1}^{K} \hat{\mu}_{c_k} \right) + \sqrt{2 \log(1/\delta)} \left( \sqrt{1/n} + \sqrt{1/(nK)} \right) + \lambda_c.$$

*Proof.* We apply [54] Theorem 2 to target domain $\mu_T = \mu_c$ and joint source domain $\mu_S = 1/K \sum_{k=1}^{K} \mu_{c_k}$ with empirical samples $\hat{\mu}_T = \hat{\mu}_c$ and $\hat{\mu}_S = 1/K \sum_{k=1}^{K} \hat{\mu}_{c_k}$. $\square$

## B.3 Encoder versus Prediction Network Error

Our goal is to learn a joint hypothesis $h$ over the entire domain $\mathcal{X}$ in two steps, first inferring the task $c$ and then inferring $x_{t+1}$ conditioned on $c$. Error from `DyAd` may result from either the encoder $g_\phi$ or the prediction network $f_\theta$. Our hypothesis space has the form $\{x \mapsto f_\theta(x, g_\phi(x))\}$ where $\phi$ and $\theta$ are the weights of the encoder and prediction network respectively. Let $\epsilon_{\mathcal{X}}$ be the error over the entire domain $\mathcal{X}$, that is, for all $c$. Let $\epsilon_{\mathtt{enc}}(g_\phi) = \mathbb{E}_{x \sim \mathcal{X}}(\mathcal{L}_1(g(x), g_\phi(x)))$ be the encoder error where $g \colon \mathcal{X} \to \mathcal{C}$ is the ground truth. We state a result that decomposes the final error into that attributable to the encoder and that to the forecaster.

**Proposition B.5.** *Assume* $c \mapsto f_\theta(\cdot, c)$ *is Lipschitz continuous with Lipschitz constant* $\gamma$ *uniformly in* $\theta$. *Then we bound*

$$\epsilon_{\mathcal{X}}(f_\theta(\cdot, g_\phi(\cdot))) \leq \gamma \epsilon_{\mathtt{enc}}(g_\phi) + \mathbb{E}_{c \sim \mathcal{C}} \left[ \epsilon_c(f_\theta(x, c)) \right] \tag{4}$$

*where the first term is the error due to the encoder incorrectly identifying the task and the second term is the error due the prediction network alone.*

The hypothesis in the second term consists of the prediction network combined with the ground truth task label $x \mapsto f_\theta(x, g(x))$. In practice, the Lipschitz constant $\gamma$ of the encoder may be indirectly minimized through weight decay.

*Proof.* By the triangle inequality and linearity of expectation,

$$
\begin{aligned}
\epsilon_{\mathcal{X}}(f_\theta(\cdot, g_\phi(\cdot))) = \mathbb{E}_{c \sim \mathcal{C}} \left[ \mathbb{E}_{x \sim \mu_c} \left[ \| f_\theta(x, g_\phi(x)) - f(x) \|_{\mathcal{Y}} \right] \right] \\
\leq \mathbb{E}_{c \sim \mathcal{C}} \left[ \mathbb{E}_{x \sim \mu_c} \left[ \| f_\theta(x, g_\phi(x)) - f_\theta(x, c) \|_{\mathcal{Y}} \right] \right] \\
+ \mathbb{E}_{c \sim \mathcal{C}} \left[ \mathbb{E}_{x \sim \mu_c} \left[ \| f_\theta(x, c) \|_{\mathcal{Y}} - \| f(x) \|_{\mathcal{Y}} \right] \right].
\end{aligned}
$$

By Lipschitz continuity,

$$
\leq \mathbb{E}_{c \sim \mathcal{C}} \left[ \mathbb{E}_{x \sim \mu_c} \left[ \gamma \| g_\phi(x) - c \|_{\mathcal{C}} \right] \right] + \mathbb{E}_{c \sim \mathcal{C}} \left[ \mathbb{E}_{x \sim \mu_c} \left[ \| f_\theta(x, c) \|_{\mathcal{Y}} - \| f(x) \|_{\mathcal{Y}} \right] \right],
$$

which, since $g(x) = c$ and by linearity of expectation,

$$
= \gamma \mathbb{E}_{c \sim \mathcal{C}} \left[ \mathbb{E}_{x \sim \mu_c} \left[ \| g_\phi(x) - g(x) \|_{\mathcal{C}} \right] \right] + \mathbb{E}_{c \sim \mathcal{C}} \left[ \mathbb{E}_{x \sim \mu_c} \left[ \| f_\theta(x, c) \|_{\mathcal{Y}} - \| f(x) \|_{\mathcal{Y}} \right] \right]
$$

by definition of $\epsilon_{\text{enc}}$ and $\epsilon_c, = \gamma \epsilon_{\text{enc}}(g_\phi) + \mathbb{E}_{c \sim \mathcal{C}} \left[ \epsilon_c(f_\theta(x, c)) \right]$. $\qquad \square$

By combining Theorem B.1, Proposition B.5, and Theorem B.4, we can bound the generalization error in terms of the empirical error of the prediction network on the source domains, the Wasserstein distance between the source and target domains, and the empirical error of the encoder. Let $\mathcal{G} = \{ g_\phi \colon \mathcal{X} \to \mathcal{C} \}$ be the task encoder hypothesis space. Denote the empirical risk of the encoder $g_\phi$ with respect to $\mathbf{X}$ by $\hat{\epsilon}_{\text{enc}}(g_\phi)$.

**Proposition B.6.** *Assuming the hypotheses of Theorem B.1, Proposition B.5, and Theorem B.4,*

$$
\begin{aligned}
\epsilon_{\mathcal{X}}(f_\theta(\cdot, g_\phi(\cdot))) \leq \gamma \hat{\epsilon}_{\text{enc}}(g_\phi) + \frac{1}{K} \sum_{k=1}^{K} \hat{\epsilon}_{c_k}(f_\theta(\cdot, c_k)) + 2\gamma R(\mathcal{G}) + 2R(\mathcal{F}) \\
+ (\gamma + 1) \sqrt{\frac{\log(1/\delta)}{2nK}} + \sqrt{2 \log(1/\delta)} \left( \sqrt{1/n} + \sqrt{1/(nK)} \right) \\
+ \mathbb{E}_{c \sim \mathcal{C}} \left[ W_1 \left( \hat{\mu}_c, \frac{1}{K} \sum_{k=1}^{K} \hat{\mu}_{c_k} \right) + \lambda_c \right].
\end{aligned}
$$

*Proof.* We provide an upper bound for the right-hand side of (4). By Theorem B.1 or [44], Theorem 3.3, we can bound

$$
\epsilon_{\text{enc}}(g_\phi) \leq \hat{\epsilon}_{\text{enc}}(g_\phi) + 2R(\mathcal{G}) + \sqrt{\frac{\log(1/\delta)}{2nK}}. \tag{5}
$$

In order to apply Theorem B.1 to the risk $\epsilon_c$ and relate it to the empirical risk, we first relate the error on the target domain back to the source domain of our empirical samples. By Theorem B.4,

$$
\begin{aligned}
\epsilon_c(f_\theta(\cdot, c)) \leq \frac{1}{K} \sum_{k=1}^{K} \epsilon_{c_k}(f_\theta(\cdot, c_k)) + W_1 \left( \hat{\mu}_c, \frac{1}{K} \sum_{k=1}^{K} \hat{\mu}_{c_k} \right) \\
+ \sqrt{2 \log(1/\delta)} \left( \sqrt{1/n} + \sqrt{1/(nK)} \right) + \lambda_c.
\end{aligned} \tag{6}
$$

Applying Theorem B.1, this is

$$
\begin{aligned}
\leq \frac{1}{K} \sum_{k=1}^{K} \hat{\epsilon}_{c_k}(f_\theta(\cdot, c_k)) + 2R(\mathcal{F}) + \sqrt{\frac{\log 1/\delta}{2nK}} + W_1 \left( \hat{\mu}_c, \frac{1}{K} \sum_{k=1}^{K} \hat{\mu}_{c_k} \right) \\
+ \sqrt{2 \log(1/\delta)} \left( \sqrt{1/n} + \sqrt{1/(nK)} \right) + \lambda_c.
\end{aligned} \tag{7}
$$

Substituting (5) and (7) into (4) gives

$$
\begin{aligned}
\epsilon_{\mathcal{X}}(f_\theta(\cdot, g_\phi(\cdot))) \leq \gamma \left( \hat{\epsilon}_{\text{enc}}(g_\phi) + 2R(\mathcal{G}) + \sqrt{\frac{\log(1/\delta)}{2nK}} \right) \\
+ \mathbb{E}_{c \sim \mathcal{C}} \left[ \frac{1}{K} \sum_{k=1}^{K} \hat{\epsilon}_{c_k}(f_\theta(\cdot, c_k)) + 2R(\mathcal{F}) + \sqrt{\frac{\log 1/\delta}{2nK}} \right. \\
\left. + W_1 \left( \hat{\mu}_c, \frac{1}{K} \sum_{k=1}^{K} \hat{\mu}_{c_k} \right) + \sqrt{2 \log(1/\delta)} \left( \sqrt{1/n} + \sqrt{1/(nK)} \right) + \lambda_c \right].
\end{aligned}
$$

Using linearity of the expectation over $c$, removing it where there is no dependence on $c$, and rearranging terms. □

Although this result does not settle either the question of end-to-end versus pre-training or encoder-forecaster versus monolithic model, it quantifies the trade-offs these choices depend on. We empirically consider both questions in the experiments section.