# OpenReview forum: "Meta-Learning Dynamics Forecasting Using Task Inference"
_NeurIPS.cc/2022/Conference — NeurIPS 2022 Accept_

### Official Review · Reviewer_A5Th · 2022-07-08

**Rating:** 7
**Confidence:** 3
**Soundness:** 4 excellent
**Presentation:** 4 excellent
**Contribution:** 3 good

**Summary:**

A physics informed meta-learning architecture is introduced to model different kinds of dynamics. An encoder generates a time-invariant latent state $\hat{z}$ representing physical properties of the observed dynamics. In combination with the dynamics field, a decoder takes this latent state to condition its prediction of the next dynamic steps.

**Questions:**

- The input and output sequences have different indices compared to the text (lines 71 and 72). It might help readers if the indices were consistent, i.e., changing $w_{1:t}$ and $\hat{w_{t+1:t+k}}$ into $w_{t-l+1:t}$ and $\hat{w_{t+1:t+h}}$.
- According to your reported changes in $\hat{c}$ when shifting the input (over time), how much does $\hat{c}$ actually change (in particular since you enforce it to stay constant with the second term in equation (1))? It would be great if you could show that $\hat{c}$ is fairly constant for one time series encoded at different temporal samples. Certainly, though, $\hat{c}$ should change meaningfully when another task is encoded.
- Since the decoder conditions its prediction on the hidden features, could you also formulate $f_c = f(x|z_c)$ to emphasize this conditioning?
- Could you use something different from $\mu$ for the data distribution in line 84, since you also choose $\mu$ when defining the mean of channels for AdaIN (line 155)?
- In line 156, should it be "For each AdaIN *channel*, a particular style is computed $s_i = $...", or are the styles indeed layer-wise? From figure 2 I assume that you only use one AdaIN layer, though, meaning you only use one style?
- Have you tested recurrent modules in the encoder and forecaster?
- How do you explain the performance of the big U-Net as forecaster compared to the small one? And do you have an intuition why U-Nets seem better on the SST data while ResNet is superior in Turbulent Flows and Ocean Currents?

**Limitations:**

Limitations not addressed by the authors. I'd be curious to learn about situations where this model might have difficulties.

**Strengths And Weaknesses:**

### Originality
Strengths:
- The AdaPad layer looks intriguing, as it allows the boundary condition (BC) to depend on $x$. In contrast to traditional padding methods (zero/constant or mirror), AdaPad thus seems able to account for dynamic BCs, such as Neumann and Cauchy. However, it might help if it additionally receives $x$ as input. Have the authors experimented with this?
- Fairly simple but highly efficient and well designed architecture to model different kinds of dynamics with one and the same forecaster

### Quality:
Strengths:
- Overall very detailed and well motivated model architecture, data description and evaluation
- Clear demonstration of successful encoding of physically relevant factors (figure 6)
- Exhaustive ablations showing the relevance of the different introduced network components

Weaknesses:
- Apart from the ablations in table 2 and in the appendix, it would be informative to see the effect of different choices of $m$

### Clarity and significance:
Strengths:
- Superiority of the model shown on both synthetic and real-world datasets

Weaknesses:
- I'm not fully clear about the conclusion of the proof and would appreciate an intuition about the result and what it actually means.
- AdaPad operator not explicitly evaluated on changing boundary conditions

---

> ### Author Response · Authors · 2022-08-01
> **Response to Reviewer A5Th - Part One**
>
> Thanks so much for your positive comments and insightful questions.
>
> $\newline$
>
> $\textbf{Q1:}$ The AdaPad layer looks intriguing... However, it might help if it additionally receives x as input. Have the authors experimented with this?
>
> $\textbf{A1:}$ That’s a good suggestion. We tried this before and it didn’t help improve the model’s prediction performance. The hidden feature z_c contains enough information for encoding the boundary conditions.
>
> $\newline$
>
> $\textbf{Q2:}$ I'm not fully clear about the conclusion of the proof...
>
> $\textbf{A2:}$ The theoretical analysis formalizes our model architecture design and learning algorithm. It informs three aspects of our approach: (1) inferring and training multiple tasks simultaneously has better generalization than learning each task independently; (2) our two-stage training scheme is motivated by the decomposition of domain adaptation error and can be used to learn a heterogeneous domain.  (3) The error of the model can be controlled by the empirical errors of the encoder and forecaster, as well as the task similarity. We are measuring task similarity with Wassertein distance in the space of observations. In short, Prop 2.1 helps us understand the trade-offs w.r.t. different terms in the error bound.
>
> In practice, we can compute the Wasserstein distance between the source and target domains as well as the empirical error of the encoder and the forecaster on the source domains in the upper bound. But it’s almost impossible to compute the Rademacher complexity. Theoretically the Rademacher complexity of the function class decreases as sample size increases and grows with the neural nets capacity. The bound tells us that we need to find the sweet spot that minimizes the sum of Rademacher complexity and empirical error.
>
> $\newline$
>
> $\textbf{Q3:}$ AdaPad operator not explicitly evaluated on changing boundary conditions.
>
> $\textbf{A3:}$ That’s a good point. We will evaluate our model explicitly on changing boundary conditions in future work. But AdaPad should have the potential to learn changing boundary conditions since it takes a sequence of historic observations as input. And for the experiments on ocean dynamics, the models are trained on data from 64*64 subregions, the boundary conditions of which can be considered open and changing.
>
> $\newline$
>
> $\textbf{Q4:}$ The input and output sequences have different indices compared to the text...
>
> $\textbf{A4:}$ We have fixed it.
>
> $\newline$
>
> $\textbf{Q5:}$ ... how much does c^actually change (in particular since you enforce it to stay constant with the second term in equation...
>
> $\textbf{A5:}$ Great suggestion! We performed the test and verified that our design indeed results in time-invariance (low time-shift error). The scaled time-shift errors with and without time-shift invariant loss averaged over all samples on the turbulent flow dataset are shown below.
> |                         | With time-shift loss     | W/O time-shift loss |
> |:-----------------------:|:------------------------:|:-------------------:|
> | Scaled Time shift error |       6.67e-10           |       1.56e-05      |
>
> $\newline$
>
> $\textbf{Q6:}$ Since the decoder conditions its prediction on the hidden features, could you also formulate  fc=f(x|zc) to emphasize this conditioning?
>
> $\textbf{A6:}$ Here we use f to refer to the forecaster, not a probability distribution. y= f(x, z_c) simply means the forecaster takes x and z_c as input.
>
> $\newline$
>
> $\textbf{Q7:}$ In line 156... For each AdaIN channel, a particular style is computed si=...", or are the styles indeed layer-wise?...
>
> $\textbf{A7:}$ In the forecaster, we apply AdaIN after every two convolutional layers (i.e. one resnet block). There is a “x N”  in the forecaster network module in figure 2, meaning we use N styles.
>
> $\newline$
>
> $\textbf{Q8:}$ Have you tested recurrent modules in the encoder and forecaster?
>
> $\textbf{A8:}$  There are two reasons that we favor convolutional modules over recurrent modules for the encoder and forecaster. One is that PredRNN (one of the baselines) with recurrent modules performs poorly. The other reason is computational efficiency. Recurrent modules need to process frames sequentially and thus are very slow when the input length and the forecasting horizon is big. This contradicts the goal of speeding up dynamics simulation with neural surrogate models.

---

> > ### Author Response · Authors · 2022-08-01
> > **Response to Reviewer A5Th - Part Two**
> >
> > $\textbf{Q9:}$ How do you explain the performance of the big U-Net as forecaster compared to the small one?
> >
> > $\textbf{A9:}$ We experimented with a bigger DyAd+Unet that has about the same number of parameters as DyAd+ResNet has to show that the DyAd+UNet was not underfitting the training data and bigger capacity does not bring better prediction performance on all datasets.
> >
> > $\newline$
> >
> > $\textbf{Q10:}$ And do you have an intuition why U-Nets seem better on the SST data while ResNet is superior in Turbulent Flows and Ocean Currents?
> >
> > $\textbf{A10:}$ This is a very interesting observation. SST data is much smoother and less turbulent than turbulent flow and ocean currents datasets. U-net contains efficient downsampling and upsampling components, which is well-suited at capturing large-scale evolution in the SST. But the downsampling part may result in loss of information so it has trouble capturing the fast-changing small eddies in the turbulence and ocean currents. A relevant study [1] also found U-net is better at predicting SST.
> > Moreover, U-net has downsampling and feature extraction to form thicker features, the padding of which are not necessarily related to the boundary conditions. It may be that AdaPad is less appropriate in this highly downsampled regime and Unet does not benefit from AdaPad as much as ResNet.
> >
> > [1] Bezenac et al; Deep Learning for Physical Processes: Incorporating Prior Scientific Knowledge.

---

### Official Review · Reviewer_xxYt · 2022-07-10

**Rating:** 7
**Confidence:** 4
**Soundness:** 3 good
**Presentation:** 4 excellent
**Contribution:** 3 good

**Summary:**

The paper proposes a method to predict dynamical systems when system parameters can differ between training and evaluation.
First, a time invariant network is trained to predict the dynamical "invariants" which could be the number of vortices in a flow or system parameters.
Then a prediction network is trained to forecast the system dynamics, taking the dynamic invariants as an input.


**Questions:**

1. How do meta-training and meta-testing sets differ? Can you quantify how large the shift is between training and testing dataset (e.g. a shift in the mean number of vortices)?  In particular, do you evaluate on values of the invariant that are not included in the training set?
2. Feeding the invariant as an input to the prediction model is one way to incorporate the information. If one can compute invariants in a differentiable way one could also incorporate this as a loss where context and predictions should not differ in the invariant.
Have you considered this aswell? It probably only works for some invariants, but may be interesting.

**Limitations:**

-

**Strengths And Weaknesses:**

### Strengths
- Interesting and innovative idea. (I may not be aware of prior work using this approach)
- Good experimental results, although maybe a bit limited datasets.
- Good and clear presentation

### Weaknesses
One potential weakness of the method is that one needs access to the invariants for training.

---

> ### Author Response · Authors · 2022-08-01
> **Response to Reviewer xxYt**
>
> Thank you so much for your positive comments and insightful questions.
>
> $\newline$
>
> $\textbf{Q1.1:}$ How do meta-training and meta-testing sets differ?
>
> $\textbf{A1.1:}$ The turbulence dataset contains 25 simulations (i.e tasks) of length 500, and both the ocean current and temperature datasets contain 25 sequences (i.e tasks) of length 500 from 25 64x64 subregions. And we use a sliding window approach to generate samples of sequences. On each dataset, we use samples from time step 1 to 400 from 20 randomly chosen tasks for training and validation.
>
> We consider two testing scenarios. For test-future, the meta-testing set contains the samples from time step 400 to 500 from the 20 chosen tasks. For test-domain, meta-testing set contains the samples from time step 0 to 100 from the remaining 5 tasks.
>
> Also note that our model is a model-based meta-learning model that does not need support sets from new tasks (in other words, our model is doing zero-shot learning). You may also see our code in the supplementary material.
>
> $\newline$
>
> $\textbf{Q1.2:}$ Can you quantify how large the shift is between training and testing dataset (e.g. a shift in the mean number of vortices)?
>
> $\textbf{A1.2:}$ Yes, We used Wassertain distance to quantify the distributional shifts between training and testing in our theoretical analysis. We also added empirical measurements of the pairwise approximate Wasserstein distance between tasks in the Figure 8 in the appendix of the updated version.
>
> $\newline$
>
> $\textbf{Q1.3:}$ In particular, do you evaluate on values of the invariant that are not included in the training set?
>
> $\textbf{A1.3:}$ Yes, the train set and test-domain set use different values of the invariant c, showing our model can generalize to new values of c not seen during training. Please see the answer to your first question.
>
> $\newline$
>
> $\textbf{Q2:}$ ...If one can compute invariants in a differentiable way one could also incorporate this as a loss where context and predictions should not differ in the invariant. Have you considered this as well? ...
>
> $\textbf{A2:}$ That's a great suggestion. We think two of our baselines, ResNet-c and Unet-c, conform to your idea.  ResNet/Unet-c are ResNet and Unet with an additional final layer that generates task parameter (invariants) c to make sure the predictions are consistent with the invariants. The models are trained directly with weak-supervision of invariants and forecasting loss altogether.

---

### Official Review · Reviewer_Nf4r · 2022-07-11

**Rating:** 8
**Confidence:** 3
**Soundness:** 4 excellent
**Presentation:** 4 excellent
**Contribution:** 4 excellent

**Summary:**

This paper proposes a meta-learning mechanism to address current generalization limitations in dynamics forecasting works, in which the majority of works learn deep learning models capable of capturing one dynamics at a time. To this end, they propose *DyAd*, a two-staged approach in which an encoder learns the time-invariant task-specific hidden features of a given observation sequence and influences a forecasting network that aims to learn the shared dynamics of the heterogeneous domain. A time-invariant loss function for the encoder is proposed to encourage time-task disentanglement between encoder and forecaster, leveraging weak-supervision from task-specific parameters. They leverage style-transfer adaptive instance normalization for forecaster adaptation and propose a novel encoder-influenced padding layer called *AdaPad* to address unknown boundary errors. Theoretical proofs are provided to quantify loss term trade-offs in the generalization error and show that there is a relationship between error and task relatedness for the source and target domains. Ablations are provided for each proposed model component and are evaluated on three physical dynamics tasks, including synthetic flows and real-world sea surface temperature and ocean current tasks.

**Questions:**

**Suggestions:**
- Currently it is hard to interpret the differences in performance of each method through the metrics alone, given that only snapshots of time are shown within the figures. Adding in a figure in the Appendix showcasing full sequence predictions would help the reader understand the quantitative to qualitative mapping for RMSE and ESE.
- The discussion on the Forecaster training and its loss function is a bit short-handed compared to other sections and it can be a bit difficult to interpret how it is exactly formulated. The Appendix highlights that multiple unrolled prediction steps are used in backpropagation while the Forecaster Training section highlights the loss is accumulated over 'different time steps'. Clarification on whether this refers to 1) each training sequence has multiple time windows evaluated on for its pure next step prediction and this is what is accumulated in loss for backpropagation or whether 2) for each window used, multiple steps are predicted out autoregressively and the multi-step sequence is used in the loss function.

**Limitations:**

The authors properly address limitations in the claims of their proofs and what information should be gleaned from them. Additionally, limitations in the metrics used to support evaluation are discussed in the Appendix. Potential negative societal impact are not discussed in the work and is denoted as such in the Author's Checklist.

**Strengths And Weaknesses:**

**Strengths:**
- This work tackles a novel task for dynamic systems through the application of meta-learning to enable learning a set of dynamics functions with shared underlying mechanics under one model. It provides ample discussion on common failings of learning dynamical systems and proposes fixes to address them. It is placed well within relevant literature and indeed tackles an insofar uncommon setting.
- The architecture choice of a task-feature encoder that adapts a forecasting network per task via inter-layer controlled padding is intuitive and the rigorous ablations to support the addition of each component strengthens the work.
- Ample baselines are provided from classical video prediction models to applied meta-learning methods used as comparisons.
- The provided code in the Supplementary Material is clean to run and has reproducible results regarding baselines and proposed model.
- The writing and presentation is clear with little confusion on details concerning architecture or the training setting. The network components are explained well and effectively leverages visualizations in its presentation.

**Minor Code Clean-up:**

README.md:
- A note regarding the base size of the dataset (>150GB) from 'data_generation.py' and how to generate smaller testing versions would help for more approachable evaluation of the provided code.
- There are a variety of unused imports throughout all of the provided scripts which should be cleaned up to reduce unneeded dependencies.

data_generation.py:
- Has mkdir errors given no path checking for output folders on subsequent runs
- Requires a local module import for phi.geom Sphere not already included
- Variable "Resolution" is undefined and needs removing

**Minor Writing Clean-up:**
- Line 623: 'backpropogation'

---

> ### Author Response · Authors · 2022-08-01
> **Response to Reviewer Nf4r**
>
> Thank you so much for your positive comments. We also really appreciate that you ran our code and reproduced our results. We’ll definitely make the code cleaner based on your suggestions.
>
> $\newline$
>
> $\textbf{Q1:}$ ... Adding in a figure in the Appendix showcasing full sequence predictions would help the reader understand ...
>
> $\textbf{A1:}$ We have added the requested visualization of full 20-step predictions of each method in Figure 7 in the appendix of the updated version.
>
> $\newline$
>
> $\textbf{Q2:}$ ... Clarification on whether this refers to 1) each training sequence has multiple time windows evaluated on for its pure next step prediction ... 2) for each window used, multiple steps are predicted out autoregressive ...
>
> $\textbf{A2:}$  This refers to 2) “multiple steps are predicted out autoregressively”. On each training sample/subsequence, the forecaster is rolled out to make multiple steps ahead predictions, so the loss for backpropagation is computed based on multiple-step prediction errors. And the number of steps is a hyperparameter we tuned (1~ 6) in our experiments

---

### Author Response · Authors · 2022-08-04
**General Response**

We would like to thank all reviewers for thoroughly reading our paper and providing very positive and high-quality feedback. We greatly appreciate all reviewers pointing out that our method is novel and well-motivated and the writing is clear. Reviewer Nf4r noted that our code is clean to run and has reproducible results including the baselines and proposed model. Both Reviewer A5Th and Reviewer xxYt pointed out that we have good experiments on both synthetic and real-world datasets. Reviewer A5Th really likes our AdaPad design for boundary conditions.

We address the individual questions of each reviewer below in specific replies.

---

### Meta-Review · Area_Chair_qj5W · 2022-08-28

**Recommendation:** Accept
**Confidence:** Certain

**Metareview:**

This work proposes a model-based meta-learning method to forecast physical dynamics. The proposed approach is able to generalize across heterogeneous domains as demonstrated in convincing sets of experiments. The reviewers found the work to be well motivated, clear and self-contained. Authors justified the proposed model architecture and the ablation studies conducted showed the importance of the network components. The authors also provided an adequate description of the data and the evaluation strategy, as well as theoretical guarantees on the generalization error in several settings.


**Award:**

Yes

---

### Decision · Program_Chairs · 2022-09-14

Accept